# Structural, Magnetic and Electronic Properties of 3d Transition-Metal Atoms Adsorbed Monolayer BC_2_N: A First-principles Study

**DOI:** 10.3390/ma12101601

**Published:** 2019-05-16

**Authors:** Feng Chen, Li Fan, Xun Hou, Chunmei Li, Zhi-Qian Chen

**Affiliations:** School of Materials and Energy, Southwest University, Chongqing 400715, China; cf9941110@email.swu.edu.cn (F.C.); fanfanli@email.swu.edu.cn (L.F.); hou_xunyx@163.com (X.H.); lcm1998@swu.edu.cn (C.L.)

**Keywords:** monolayer BC_2_N, transition metal atom, adsorption, magnetism

## Abstract

Based on the monolayer BC_2_N structure, the structural, electronic and magnetic properties of 3d transition metal (TM) atoms (V, Cr, Mn, Fe, Co and Ni) adsorbed on the monolayer BC_2_N, are studied by using the first principle method. The results show that 3d transition metal atoms are stably adsorbed on the monolayer BC_2_N. The most stable adsorption sites for V, Cr, and Mn atoms are the hollow adsorption site (H) of BC_2_N, while the other 3d TM atoms (Fe, Co, Ni) are more readily adsorbed above the C atoms (Tc). The majority of TM atoms are chemically adsorbed on BC_2_N, whereas Cr and Mn atoms are physically adsorbed on BC_2_N. Except for Ni, most 3d transition metal atoms can induce the monolayer BC_2_N magnetization, and the spin-charge density indicated that the magnetic moments of the adsorption systems are mainly concentrated on the TM atoms. Moreover, the introduction of TM atoms can modulate the electronic structure of a single layer of BC_2_N, making it advantageous for spintronic applications, and for the development of magnetic nanostructures.

## 1. Introduction

Because of their peculiar electronic properties, two-dimensional materials have many excellent properties that are different from three-dimensional materials [1,2,3]. In 2004, Novoselov and Geim successfully obtained graphene by the mechanical stripping method [4]. The discovery of graphene opens the way for the study of two-dimensional materials. Subsequently, graphene-like materials, such as two-dimensional hexagonal boron nitride (h-BN), were successfully prepared [5], and its properties were studied theoretically and experimentally [6,7,8]. The spatial structure of monolayer h-BN and graphene was found, they are very similar, but their electronic structure is very different [9]. So researchers have a lot of interest in B–C–N, which is a new compound that can be formed by combining the two compounds. Since the B–C–N layered material family has been synthesized by chemical vapor deposition (CVD) [10,11], scientists have begun to study B–C–N materials theoretically and experimentally. The early studies on B–C–N ternary compounds were mainly focused on BC_2_N compounds. In 1989, Liu and Cohen et al. put forward the monolayer BC_2_N structure model by first-principles calculation for the first time [12]. Subsequently, Nozaki and Itoh et al. researched the structure and properties of monolayer BC_2_N by the semiclassical method [13]. In experiments, Qin Li et al. prepared hexagonal B–C–N nanocrystalline films by chemical vapor deposition [14]. By adjusting the experimental parameters, the ratios of the B, C and N elements could be adjusted, thus the BC_2_N compounds were prepared [15].

Due to the application prospect of two-dimensional materials in spin electronic devices and nanoelectronics, the magnetic properties of layered materials have attracted much attention. In order to realize their practical application, it is very important to modulate their electronic structure and magnetic properties [16,17,18]. In recent years, research on the magnetic properties of the system by modulating two-dimensional materials has been paid more and more attention. Some theoretical studies have shown that defects can cause magnetic phenomena in BC_2_N [19]. It is found that the structure with vacancy defects has a higher formation energy and local spin magnetic moment than the system with inverse defects. Interestingly, a large number of studies [20,21,22,23] on graphene and h-BN have shown that transition metal atoms can also magnetize graphene and monolayer hexagonal boron nitride, resulting in strange spin-poles in the adsorption system. The electronic structure is modulated. There are many ways of modulation, such as adsorption, doping, applying stress and applying an electric field, etc., among which adsorption is an effective method [24,25,26,27,28,29,30]. 

Two-dimensional BC_2_N combines the advantages of graphene and h-BN [31], with higher carrier mobility and a certain band gap, but the lack of magnetism limits its use in nanoelectronics and spintronics. Therefore, in order to enable single-layer BC_2_N to be used in spintronics and magnetic nanostructures, it is particularly important to use TM atoms to modulate the electrons and magnetism of two-dimensional BC_2_N. At present, there is no theoretical research on the two-dimensional BC_2_N adsorbed by TM atoms. Therefore, it is necessary to theoretically explore the stability, magnetic and electronic structure of the two-dimensional BC_2_N modulated by these TM atoms.

For this, the transition metal atom is considered to adsorb on the monolayer BC_2_N, and the calculation is done in the case of spin polarization to study the stability structure, magnetic properties and electronic properties of the adsorption system. It has been found that the transition metal atom can be stably adsorbed on the two-dimensional BC_2_N. Except for Ni, the TM atoms can magnetize the two-dimensional BC_2_N. 

## 2. Calculation Method and Structural Models 

The calculations implemented with CASTEP code based on density functional theory (DFT) [32,33,34] were performed to study the structural, magnetic and electronic properties of 3d TM atoms (V, Cr, Mn, Fe, Co and Ni) adsorbed on the monolayer BC_2_N. The interelectron exchange correlation potential was used to describe the phase between electrons and ions by using Perdew–Wang 91 (PW91) [35] under the generalized gradient approximation (GGA). In order to avoid the interaction of adjacent TM atoms, we need to set the appropriate supercell size. The change of adsorption energy of the 5 × 5 supercell and the 4 × 4 supercell is less than 1%, so it indicates that the 4 × 4 supercell is large enough to avoid the interaction of adjacent TM atoms.

First, we completely optimize the 4 × 4 supercell of BC_2_N with the plane wave truncation energy is 500 eV. The Brillouin region integral of pure BC_2_N adopts the special k-point sampling method of 7 × 7 × 1 monkhost-pack to ensure the accuracy of the results [36], and the special k-point sampling of the monkhost-pack of the adsorption system is increased to 10 × 10 × 1. A 15Å vacuum space in the Z direction was applied to each side of the supercell, in order to avoid the interactions between the individual layers. Grimme’s semi-empirical dispersion correction (DFT+D) method is added to consider van der Waals interaction [37]. In all processes of self-consistent calculation, the convergence criterion follows that the force on each atom is less than 0.05 eV/Å, and the convergence threshold is 10^−6^ eV. The coulomb interaction (GGA+U) was introduced to describe the d electrons of the TM atom [38].

The lattice constants and binding energies of two-dimensional BC_2_N optimized by PW91 are 4.92, −8.821 Å, respectively. These results are in agreement with previous theoretical reports [39]. After optimization, the bond lengths of the C–C bond, B–N bond, C–N bond and C–B bond are 0.142, 0.144, 0.139 and 0.152 Å, respectively. They are very consistent with the theoretical data [39], which means that our C–N bond is the most effective in this structure. Figure 1a is a schematic diagram of transition metal atoms adsorbed on two-dimensional BC_2_N. 

Five possible adsorption sites are considered, namely the H site (above the center of the hexagonal ring of BC_2_N), the T_B_ site (on top of the B atom), T_C_ site (on top of the C atom), T_N_ site (on top of the N atom), and Bri site (above the middle of our B–C bond) are showed in Figure 1b. 

## 3. Results and Discussion

BC_2_N has been structurally optimized and contains three kinds of atoms (4 carbon, 2 boron and 2 nitrogen). As shown in Figure 2a, the band structure of BC_2_N is calculated under the condition of spin polarization. M and K represent the two height symmetry points of the Brillouin zone of the supercell, that is (0, 0.5, 0) and (−1/3, 2/3, 0). The Fermi level is set to zero energy, and is represented by a horizontal black dotted line. Figure 2b is the total density of states of BC_2_N, and it can be seen that BC_2_N has no spin splitting near the Fermi level, and therefore no magnetism. Next, the lowest unoccupied conduction band (LUCB) and the highest occupied valence band (HOVB) at the G-point show that the BC_2_N is a direct band gap semiconductor with a band gap of 1.58 eV. This result is in agreement with a previous theoretical report [39]. A similar band gap can also be obtained from the total density of states.

### 3.1. Stability

We have selected six magnetic 3d TM atoms (V, Cr, Mn, Fe, Co and Ni) to adsorb on the monolayer BC_2_N, and calculate the adsorption system under the spin polarization state. In order to study the most favorable adsorption structure, the adsorption energy [20] is defined as:(1)$$E_{adsorption}=E_{\left(adatom+BC_{2}Nsheet\right)}-E_{BC_{2}Nsheet}-E_{\mathrm{adatom}}$$
where $E_{\left(\mathrm{adatom}+{\mathrm{BC}}_{2}\mathrm{Nsheet}\right)}$ is the total energy of the adsorption system. $E_{BC_{2}Nsheet}$ and $E_{\mathrm{adatom}}$ are the total energy of the isolated pure BC_2_N monolayer and total energy of the corresponding isolated metal atom, respectively. 

Figure 3 shows the adsorption energy of TM atoms at different adsorption positions of monolayer BC_2_N. The adsorption system of V, Cr and Mn has the lowest adsorption energy at the H site, while Fe, Co, and Ni have the lowest adsorption energy at the Tc site. By definition, a negative value corresponds to an exothermic adsorption. The smaller the adsorption energy, the more stable the 3d TM atom adsorption. The stronger the interaction between the TM atoms and the monolayer that BC_2_N implies, the stronger the binding strength, and as such, the structural stability can be determined by the adsorption energy. Therefore, the most stable adsorption sites for V, Cr, and Mn atoms are the hollow adsorption site (H) of BC_2_N, while the other 3d TM atoms (Fe, Co, Ni) are more readily adsorbed above the C atoms (Tc). The corresponding most stable structure of all adsorption systems is shown in Figure 4. The relative stability of these positions is judged by comparing their adsorption energies $\left(E_{a}\right)$. Table 1 lists the adsorption position, adsorption energy ($E_{a})$ and the perpendicular distance between TM atoms and the monolayer BC_2_N (*d_TM-h_*). Except for the adsorption system of Cr and Mn, the adsorption energy of other TM atoms (V, Fe, Co, Ni) adsorbed on monolayer BC_2_N is quite large, ranging from −1.574 to −0.582 eV. Obviously, their adsorption energy is greater than 0.5 eV with lower perpendicular distance, which is chemical adsorption, while the adsorption energy of Cr and Mn is less than 0.5 eV with higher perpendicular distance, which belongs to physical adsorption. We know that the 3d orbital of free Cr, Mn atoms are half-filled. The half-filled 3d orbital weakens the interaction between transition metal atoms and the monolayer BC_2_N, reducing the adsorption energies of these systems. The scenario is similar to the adsorption of the Mn atom adsorbed on two-dimensional InSe [40].

In order to explore the different trends of adsorption energy of the adsorption atoms and monolayer BC_2_N, we calculated the charge density difference of the adsorption system. In order to clearly reveal the charge density difference, the dicing plate perpendicular to the BC_2_N plane is jointly determined by the transition metal atom and the single layer BC_2_N, as shown in Figure 5. For the H adsorption configuration, the charge density is accumulated near the transition metal atoms, and the hexagonal ring formed by B–C–N is off centered near the C atom. The charge accumulation of the Cr atom adsorption system is the smallest, and the combined strength with BC_2_N is also the smallest, so the relevant adsorption energy of Cr is the smallest. There is no obvious channel between the Cr atom and BC_2_N, that is, it is difficult for them to form a bond, so it belongs to physical adsorption. On the contrary, for Tc adsorption configurations, that is, the Fe, Co and Ni adsorption systems, the charge accumulation expands as the atomic number increases, which results in shorter bond strength and lower adsorption energy. Generally, the bond strength is directly related to the bond length of the transition metal atoms, while the shorter bond length corresponds to the stronger bond, which leads to lower adsorption energy and a more stable configuration. However, the relationship between transition metal atoms and the nearest B, C and N atoms is more complicated. In this case, the charge accumulation between the transition metal atoms and the center of the hexagonal ring or the nearest C determines the bond strength, thus it becomes a major factor affecting the energy of adsorption.

### 3.2. Magnetic Properties

On the basis of the stable adsorption of transition metal atoms on the monolayer BC_2_N structure, the magnetic properties of the adsorbed system were studied by Mulliken layout analysis. As shown in Table 2, it is evident that the Ni adsorption system exhibits zero magnetic moments, that is, the system is nonmagnetic, and the magnetic moment exhibited by Ni atoms adsorbing BC_2_N is zero, which is similar to the adsorption of Ni atoms on graphene [43]. However, the adsorption of other transition metal atoms can make the system exhibit larger magnetic moments. In the adsorption systems of V, Cr, Mn, Fe and Co, the total magnetic moments are 5.0, 6.0, 5.0, 4.0 and 1.0 *μ*_B_, respectively. The contributions of the V, Cr, Mn, Fe, Co metal atoms to the system are respectively, 4.33, 5.69, 5.32, 4.24 and 0.59 *μ*_B_. It can be concluded that transition metal atoms have the greatest contribution to the total magnetic moment of the system. The magnetic moments of V, Cr, Mn, Fe, Co and Ni atoms in the free state are 5, 6, 5, 4, 3 and 2 *μ*_B_, respectively. Their total magnetic moment is equal to that of their free state. This characteristic can be used to fabricate molecular magnets with metal covered BC_2_N.

In addition, the introduction of transition metal atoms (V, Cr, Mn, Fe, Co) makes the nonmagnetic BC_2_N produce a small amount of magnetic moment, and the B, C and N atoms near the best stable adsorption site contribute slightly to the total magnetic moment of the system. The contribution of C atoms to the total magnetic moment is 0.4, 0.06, −0.24, 0.11 and −0.03 *μ*_B_, and the contribution of B atoms is 0.16, 0.03, −0.04, −0.04 and −0.01 *μ*_B_, while the contribution of N is 0.11, 0.03, −0.07, −0.05 and −0.02 *μ*_B_. Clearly, the magnetic moment contribution is positive or negative due to the difference of the transition metal atoms introduced. For the V, Cr adsorption system, the contribution of B, C and N atoms to the total magnetic moment of the adsorption system is positive. In the adsorption system of the Mn, Fe and Co atoms, the contribution of B, C and N atoms to the total magnetic moment is almost negative. In all stable adsorption systems, C atoms contribute the most to the total magnetic moment of the adsorption system because the transition metal atoms have the most pronounced effect on the induction of C atoms in BC_2_N, and are the most coupled. 

To better understand the magnetic distribution of transition metal atoms adsorbed on BC_2_N, we use the spin charge density (SCD). Spin charge density is defined as ρr=ρ↑(r)−ρ↓ (r), where ρ↑(r) and ρ↓ (r) represent the spin-up and spin-down charge density of the BC_2_N system adsorbed by the transition metal atoms, respectively. 

As shown in Figure 6, we have calculated the spin charge density diagram of the adsorption system. It can be seen from the SCD diagram that the spin-polarized charge is localized around the transition metal atoms, which implies that the magnetic moment of the adsorption system is mainly provided by the transition metal atoms. There is no spin-polarized charge around the Ni atomic adsorption system, which indicates that the magnetic moment of the free Ni atom is annihilated when the Ni atom is adsorbed on the monolayer BC_2_N, so the magnetic moment of the system will be zero. From the SCD diagram we cannot know exactly whether the magnetic moment is related to the 3d orbital of the transition metal atom. So, the magnetic characteristics of the adsorption system will be analyzed by the density of states.

### 3.3. Electronic Structures 

To investigate how the 3d transition metal atom affects the electronic structure of a single layer of BC_2_N, the total density of states (TDOS) of all adsorption systems are shown in Figure 7. Apparently, the spin-up and spin-down TDOS of the former five adsorption systems are asymmetrically distributed, and thus it means that the TM atoms (TM=V, Cr, Mn, Fe, Co) can effectively induce the monolayer BC_2_N magnetization. The spin-polarized state is mainly located near the Fermi level. In order to analyze the spin polarization of the adsorption systems [45], the spin polarizability is defined as:(2)$$P\left(E_{\left(f\right)}\right)=\frac{D\left(E_{\left(f\right),↑}\right)-D\left(E_{\left(f\right),↓}\right)}{D\left(E_{\left(f\right),↑}\right)+D\left(E_{\left(f\right),↓}\right)}$$
where $D\left(E_{\left(f\right),↑}\right)$ and $D\left(E_{\left(f\right),↓}\right)$ are the density of states (DOS) of majority (spin up) and minority spin (spin down) at the Fermi level, respectively. In addition, in order to analyze the source of the local magnetic moment in detail, the DOS and the PDOS of the most stable adsorption system are shown in Figure 7. The positive and negative DOS represent spin up and down, respectively, and the Fermi level is set to zero level, and is represented by a vertical dashed line. According to Figure 7, it can be found that when a transition metal atom is introduced, some impurity states are generated in the band gap of the single layer BC_2_N, and these spin polarizations away from the Fermi level (less than −1.8 eV and above 1 eV) are not significant, so the magnetic source in the adsorption system is the polarization of the boundary orbit. However, for the structure of Ni adsorbed on monolayer BC_2_N, the PDOS of all orbitals is up and down. The spins are symmetrical, and the system exhibits the characteristic of zero magnetic moment. 

As shown in Figure 7a, when V is adsorbed at the most stable position on the monolayer BC_2_N, we can clearly find that there are spin-up states at the Fermi level, while the spin-down states are zero at the Fermi level. Therefore, the spin polarizability of the system reaches 100%. Meanwhile, the magnetic moment is mainly related to the state of the majority-spin channel of V near the Fermi level, and the contribution of the states below −1.8 eV to the magnetic moment is almost zero. Through the analysis of the PDOS, The main contribution of spin polarization is derived from the 3d and 4s orbitals of V, and the contribution of the 3d orbitals is greater than that of the 4s-orbitals. In addition, in the vicinity of the Fermi energy level, the 2p orbital of B, C and N also contributes a little to the spin polarization, which derives from the coupling between the 3d and 4s orbitals of V and 2p orbital of the B, C and N atoms. Compared with the contribution of the 2p orbitals of these B, C and N atoms to the V adsorbed system, the contribution of the C atom is the larger than all the others. This is because the coupling between our V atom and C atom is stronger than that between this V atom and the other atoms at the best adsorption site.

The density of states of the most stable adsorption system for Cr atoms is shown in Figure 7b. When the Cr atom is adsorbed at the H position, there is a state of spin upward at the Fermi level, and the DOS of spin down is zero, so the spin polarizability at the Fermi energy level is 100%. 

In addition, the magnetic moment of the system is mainly related to the spin upward state ranging from −1.35 eV to 0.312 eV. The main contribution of spin polarization is derived from the 3d orbital and 4s orbital of Cr, and the contribution of this 3d orbital and 4s orbital to spin polarization is equivalent, but the state at the Fermi energy level is not the biggest contribution to the magnetic moment of the system, and the most important state is in the energy range of [−1.35 eV, −0.312 eV], which is mainly provided by the 3d and 4s orbitals of Cr. At the same time, the spin upward states ranging from −1.35 eV to 0.312 eV slightly come from the 2p orbital of the B, C and N atoms, and because the 3d and 4s orbitals of Cr and the 2p orbits of the nearest neighbor B, C and N atoms are hybrid, the spin polarization of the 2p orbital of these B, C and N atoms is induced. The stronger the hybrid, the stronger the induction. The spin states of Cr and the 2p orbitals of C have the strongest hybridization. Compared with the contribution of the B and N atoms to the spin polarization, the contribution of the C atom is the greatest.

From Figure 7c, we can see that Mn is adsorbed to the H position of monolayer BC_2_N, there is no state at the Fermi energy level, and the spin down state exists, so the spin polarizability at the Fermi level is 100%. It is found that the spin polarization near the Fermi energy level is mainly contributed by the 4s orbital of Mn, and the contribution of 3d orbital of Mn is almost zero. The contribution of the state at the Fermi level to the overall magnetic moment of the system is small. The unoccupied state of the electron at 0.75 eV contributes greatly to the magnetic moment of the system, which is mainly attributed to the 3d contribution of Mn.

As shown in Figure 7d, spin-down states for the Fe adsorbed system are nonzero at the Fermi level, while the spin-up states are zero, thus it means that the spin polarizability is 100%. The spin polarization at the Fermi level is mainly contributed by the 3d orbital of Fe, and the contribution of the 4s orbital to the spin polarization is almost zero. The electron occupied states of the spin up state from −0.49 eV to −0.06 eV are mainly contributed by the 4s orbital of Fe, and the contribution of the 3d orbital of the Fe atom is almost zero. B, C and N atoms also contribute to the magnetic moment of the system, and the nearest neighbor to the adsorption site, the B, C and N atoms, contribute greatly to the spin polarization. This is due to the existence of hybridization between the 4s orbital of the Fe atom and the 2p orbital of the B, C and N atoms. The 4s orbital of the Fe atom and the 2p orbital of the C atom have the strongest hybridization, so the contribution of the C atom to spin polarization is the greatest.

Figure 7e shows the DOS of the most stable adsorption system for Co atoms. There is DOS value just in the spin-down state for the Co adsorbed system, and it means that spin polarizability at the Fermi energy level is 100%. The spin-down states at the Fermi level and the states in [−1.89 eV, −0.51 eV] play a major role in the magnetic moment of the system. The PDOS analysis shows that the spin-down state at the Fermi level is mainly contributed by the 3d orbital of Co, while the 4s orbital of Co provides a small contribution. 3d and 4s orbitals of Co are hybrid with the monolayer BC_2_N. The spin-polarized state of Co interacts with B, C and N, which induces spin polarization also induced by the monolayer BC_2_N.

Figure 8f is the DOS of the Ni atom adsorbed monolayer BC_2_N. The DOS of majority and minority are completely symmetric. Therefore, there is no spin polarization for the Ni adsorbed system. Through the analysis of the PDOS, the states of majority and minority in the energy range of [−1 eV, 0.96 eV] are mainly contributed by the 3d and 4s orbitals of Ni, and the contribution of the spin-up state is the same as that of the spin-down state. At the same time, the 3d and 4s orbitals of Ni have very strong hybridization with the 2p orbitals of the nearest adjacent B, C and N atoms, and the strong coupling of these Ni atoms with the monolayer BC_2_N makes the 2p states of B, C and N also induce symmetric spin states.

It is well known that when the adsorption system contains 3d TM atoms, the Hubbard-type Coulomb interaction between d-d electrons has a great effect on the electronic structure and Fermi level of the adsorption system. In this work, the adsorption system contains 3d TM atoms, and the electron location of 3d TM atoms is not well described in GGA, so it is necessary to explore what changes exist in the electronic structure when electronic localization is considered. To explore this issue, the Hubbard-type Coulomb interaction should be considered by calculating GGA + U [38]. Fig. 8 shows the DOS of the adsorption system with GGA+U (U = 2, 4, 6, 8) for 3d orbitals of TM atoms. When U = 0, the spin polarization of the adsorption system at the Fermi level reaches 100% (half-metallic property). It can be seen that when U changes from 2 to 8 eV, the adsorption systems (V, Co, Mn) still possess this half-metallic property, but the half-metallic properties for adsorption systems of the Fe and Cr atoms disappear. It means that when the Coulomb interaction between d electrons is considered, the half-metallic property of the adsorption system of Fe and Cr atoms changes.

## 4. Conclusions

In summary, the stable structure, magnetic and electronic properties of 3d transition metal atoms adsorbed on a 4 × 4 monolayer BC_2_N have been investigated by the first-principles method. The calculated results show that the transition metal atoms (V, Cr, Mn, Fe, Co, Ni) can stably adsorb on the monolayer BC_2_N. The study on its structure shows that the most stable adsorption sites for V, Cr and Mn atoms are located in the hollow adsorption site (H) of the BC_2_N monolayer, while other 3d TM atoms (Fe, Co, Ni) prefer to adsorb on the C atom (Tc). By studying its adsorption energy, Cr and Mn are physically adsorbed on monolayer BC_2_N, whereas the TM atoms (V, Mn, Fe, Co, Ni) are chemically adsorbed on monolayer BC_2_N. Studies on magnetic properties show that most 3d transition metal atoms can make the monolayer BC_2_N magnetization, and the spin charge density indicated that the magnetic moments of adsorption systems are mainly concentrated on the TM atoms. Through the analysis of the DOS, it can be seen that the introduction of 3d TM atoms (V, Cr, Mn, Fe, Co) leads to the spin polarization of the adsorption system, and the spin polarizability reaches 100% (half-metallic property), which has the characteristics of spin filter material. There is a hybrid phenomenon between the 3d orbital of TM atoms and the 2p orbital of its nearest neighbor B, C and N atoms. Since the 3d TM atoms have a spin-polarization characteristic, 3d orbitals can also induce auto-spin polarization of the 2p orbitals of the nearest adjacent B, C and N atoms. In addition, the Hubbard U value has an effect on these five magnetic (V, Cr, Mn, Fe, Co) adsorption systems. When U = 0, it can be found from the DOS at the Fermi level that all five configurations are semi-metallic. As the U increases, this half-metallic property for the adsorption systems of Fe and Cr atoms disappears with the conversion of half metals to metals. 

## Figures and Tables

**Figure 1 materials-12-01601-f001:**
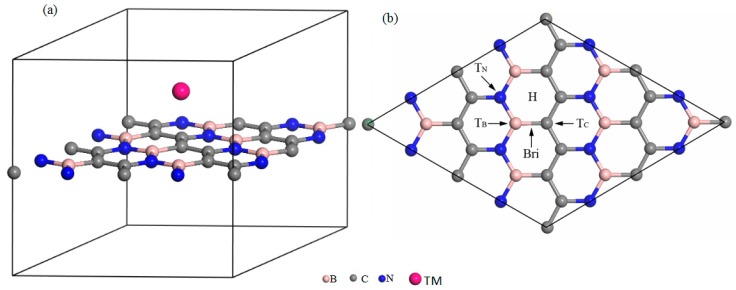
(**a**) The transition metal atom adsorption monolayer BC_2_N, (**b**) different adsorption sites of a single transition metal (TM) atom adsorbed on two-dimensional BC_2_N.

**Figure 2 materials-12-01601-f002:**
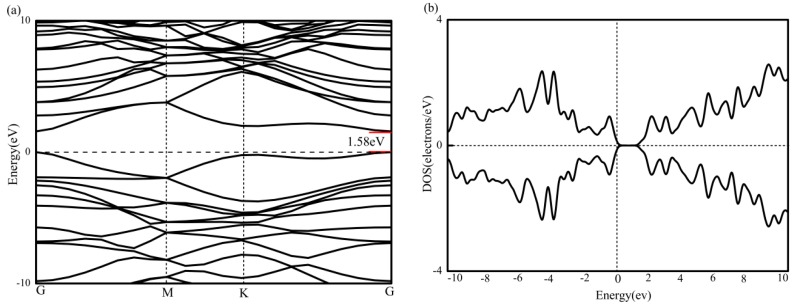
(**a**) The band graph of BC_2_N, and (**b**) The density of states of BC_2_N.

**Figure 3 materials-12-01601-f003:**
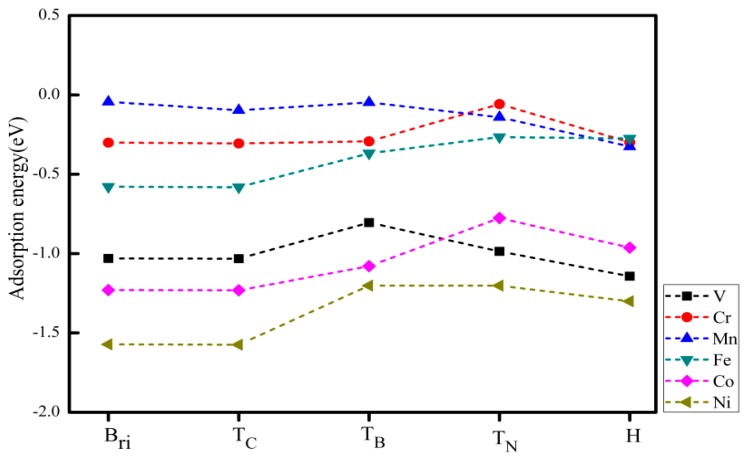
The adsorption energy of TM atoms adsorbed to different sites of single layer BC_2_N.

**Figure 4 materials-12-01601-f004:**
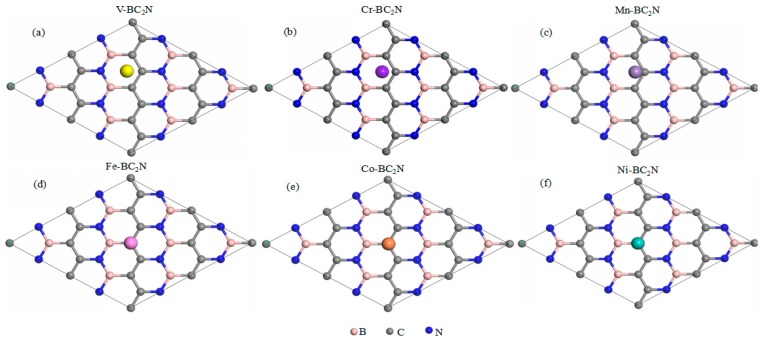
Top view of the most stable structures for transition metal atoms adsorbed monolayer BC_2_N.

**Figure 5 materials-12-01601-f005:**
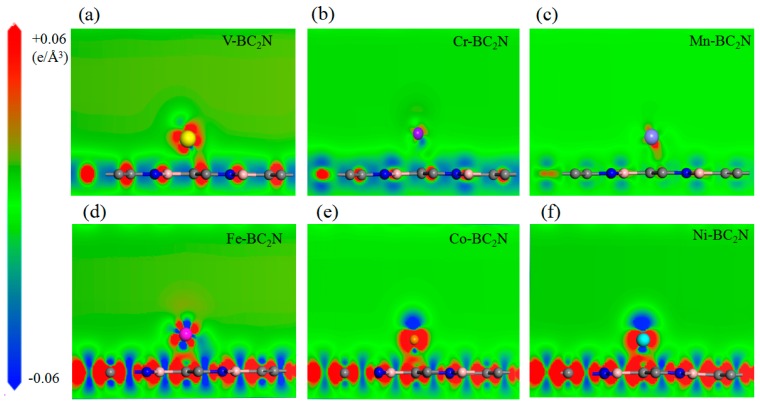
(**a**)–(**f**) The differential charge density of Ni adsorbed BC_2_N. All the adsorbed atoms are located in the stable position of geometric optimization, the loss of electrons is expressed in blue, and the accumulation of electrons is expressed in red.

**Figure 6 materials-12-01601-f006:**
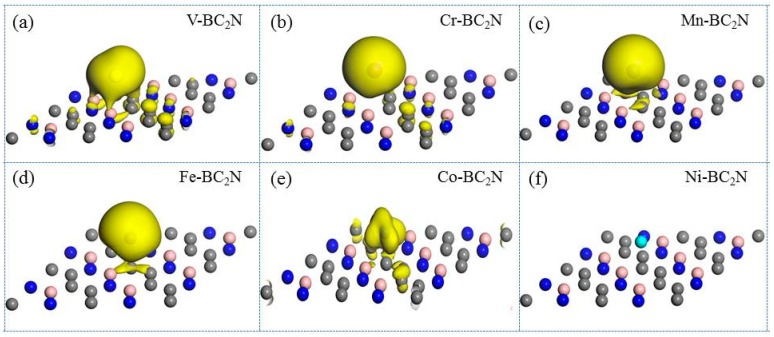
The spin charge density (SCD) of transition metal atoms adsorbed BC_2_N, and (**a**)–(**f**) represent the SCD of BC_2_N adsorbed by V, Cr, Mn, Fe, Co, and Ni, respectively. This surface value of the charge density is set to 0.003e/Å^3^.

**Figure 7 materials-12-01601-f007:**
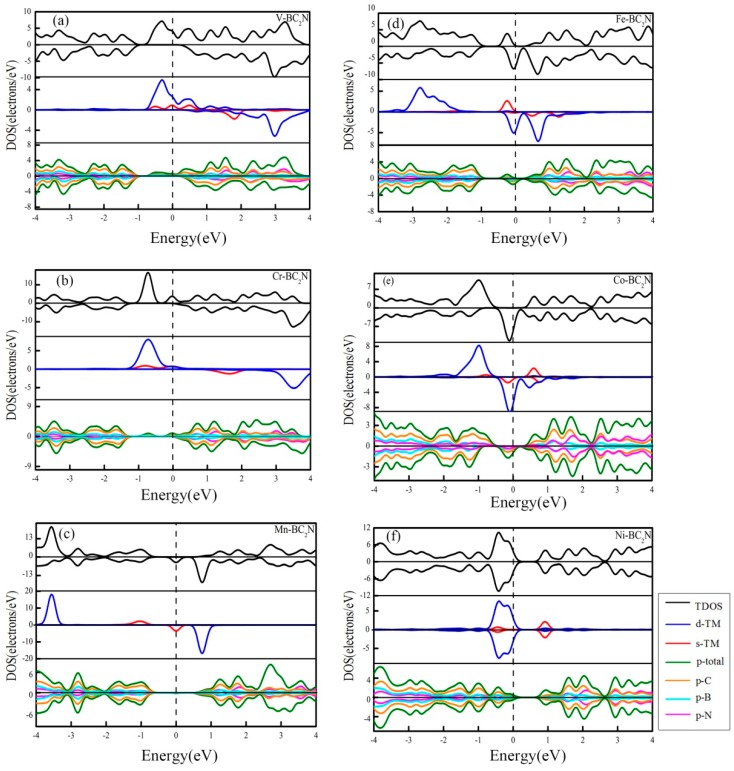
(**a**)–(**f**) The density of states for the monolayer BC_2_N doped by transition metal atoms. The upper, middle and lower sheets are the total density of states of all stable systems, the partial density of states of the 3d transition metal atoms and the partial density of states of the B, C and N atoms, respectively.

**Figure 8 materials-12-01601-f008:**
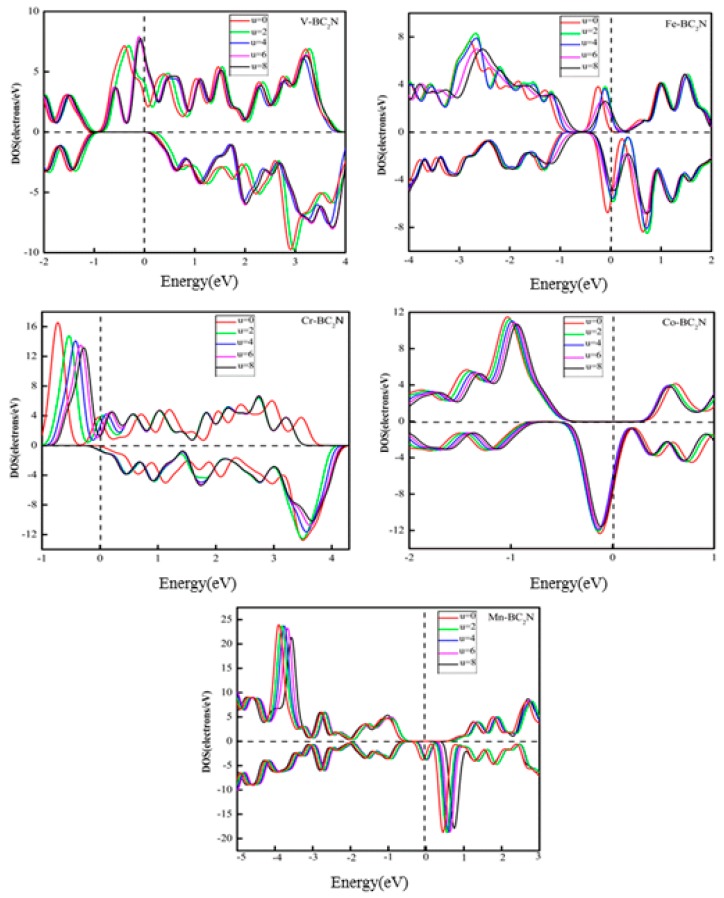
The total density of states of TM-BC_2_N (TM = V, Cr, Mn, Fe, Co, Ni) by generalized gradient approximation (GGA) and GGA+U with U= 0, 2, 4, 6, 8 eV for TM-3d electrons.

**Table 1 materials-12-01601-t001:** The adsorption position, the adsorption energy (*Ea*), the perpendicular distance between TM atoms and monolayer BC_2_N (*d_TM-h_*).

Adatom	V	Cr	Mn	Fe	Co	Ni
**site**	H	H	H	T_C_	T_C_	T_C_
***d_TM-h_* (Å)**	1.915	2.224	2.218	2.027	1.921	1.871
	1.82^a^	2.88^b^	2.83^b^	2.21^c^	2.21^d^	1.71^c^
$E_{a}$ **(ev)**	−1.143	−0.299	−0.326	−0.582	−1.131	−1.574
	−1.60^a^	−0.48^b^	−0.81^b^	−0.77^c^	−1.13^d^	−1.97^c^

^a^ Ref. [41]; ^b^ Ref. [42]; ^c^ Ref. [43]; ^d^ Ref. [44].

**Table 2 materials-12-01601-t002:** The most stable position of each metal atom of the adsorption system, the adsorption energy ($E_{a}$), the vertical height (*d_TM-h_*) between adsorption atoms and BC_2_N plane, the total magnetic moment ($\mu_{tot}$) of the adsorption system, magnetic moment of adsorption atom ($\mu_{TM}$), and atomic magnetic moment of B, C and N ($M_{i}$), ($\mu_{i}$) is the magnetic moment of the free-standing states of transition metal atoms.

Adatom	Site	*d_TM-h_* (Å)	$E_{a}(\mathrm{ev})$	$\mu_{tot}$ $\left(\mu_{B}\right)$	$\mu_{TM}$ $\left(\mu_{B}\right)$	$M_{B}\left(\mu_{B}\right)$	$M_{C}\left(\mu_{B}\right)$	$M_{N}$ $\left(\mu_{B}\right)$	$\mu_{i}\left(\mu_{B}\right)$
**V**	H	1.945	−1.143	5.0(3.0^a^)	4.33	0.16	0.34	0.11	5
**Cr**	H	2.224	−0.299	6.0(5.7^a^)	5.69	0.03	0.06	0.03	6
**Mn**	H	2.028	−0.326	5.0(3.0^a^)	5.32	−0.04	−0.24	−0.07	5
**Fe**	T_C_	2.027	−0.582	4.0(2.0^a^)	4.24	−0.04	0.11	−0.05	4
**Co**	T_C_	1.921	−1.131	1.0(1.0^a^)	0.59	−0.01	−0.03	−0.02	3
**Ni**	T_C_	1.871	−1.574						2

^a^ Ref. [40].

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
