# Peer review of "Structural, Magnetic and Electronic Properties of 3d Transition-Metal Atoms Adsorbed Monolayer BC2N: A First-principles Study"

_materials, 2019, doi:10.3390/ma12101601_

Round 1
Reviewer 1 Report
Dear authors,
all comments are included in the manuscript. In the Discussion section, you should address some papers dealing with similar thematic and compare your results with others - this part you should improve the most.
Some format and editing errors are detected.

Author Response
Dear Reviewer:
Thank you for your comments concerning our manuscript entitled “Structural, magnetic and electronic properties of 3d transition-metal atoms adsorbed monolayer BC2N: a first-principles study” (ID: materials-501745). Those comments are all valuable and very helpful for revising and improving our paper, as well as the important guiding significance to our researches. We have studied comments carefully and have made correction which we hope meet with approval. Revised portion are marked in red in the paper.
Some corrections can be found in the manuscript.
Response to Reviewer 1 Comments
Point 1: Some corrections about formatting and editing
Response 1: We have carefully read the documentation and made corresponding corrections to some of the formatting, spelling, and grammatical errors that appear in the text, and marked them in red.
Point 2: In the discussion section, some similar theories in the article should be resolved, and the data results in the paper should be compared with others.
Response 1: As the reviewers considered, we should pay attention to similar theories and comparisons with the literature data. In this regard, we made some corrections. Some similar theories in the article, such as "adsorption energy" have been cited in the literature, and the rewritten sentences have been modified and marked in red on page 4. In addition, we compare our results with those of others on page 5. References [42]-[45] mainly compare the adsorption energy and adsorption height of the same type of adsorption, we found the data to be consistent, which proves that our results are reliable.
Point 3: you could improve the quality of Fig. 6 and Fig 7.
We have improved the resolution of Fig. 6 and Fig. 7. Importing images into a word file will affect the quality of the image,so we can provide our original image separately.
Reviewer 2 Report
The manuscript titled "Structural, magnetic and electronic properties of 3d transition-metal atoms adsorbed monolayer BC2N: a first-principles study" is generally well written. However, the authors need to address the following:
review for minor grammatical errors and typos
Emphasis the novelty of this work at the end of the introduction as this is currently not very obvious
Define every acronym in their first use in the manuscript
The method is lacking. Please provide more step by step details on the theoretical methods applied and state clearly every assumption made.
Author Response
Dear Reviewer:
Thank you for your comments concerning our manuscript entitled “Structural, magnetic and electronic properties of 3d transition-metal atoms adsorbed monolayer BC2N: a first-principles study” (ID: materials-501745). Those comments are all valuable and very helpful for revising and improving our paper, as well as the important guiding significance to our researches. We have studied comments carefully and have made correction which we hope meet with approval. Revised portion are marked in red in the paper.
Some corrections can be found in the manuscript.
Response to Reviewer 2 Comments
Point 1: review for minor grammatical errors and typos
Response 1: We carefully checked and modified the grammar and typos in the article and marked them in red.
Point 2: Emphasis the novelty of this work at the end of the introduction as this is currently not very obvious
Response 2: We carefully studied the introduction part of the article and made the following changes to the penultimate paragraph to highlight the necessity and novelty of the article on page 2. “Two-dimensional BC2N combines the advantages of graphene and h-BN [31], with higher carrier mobility and a certain band gap. But the lack of magnetism limits its use in nanoelectronics and spintronics. Therefore, in order to enable single-layer BC2N to be used in spintronics and magnetic nanostructures, it is particularly important to use TM atoms to modulate the electrons and magnetism of two-dimensional BC2N. At present, there is no theoretical research on the two-dimensional BC2N adsorbed by TM atoms. Therefore, it is necessary to theoretically explore the stability, magnetic and electronic structure of the two-dimensional BC2N modulated by TM atoms.”
Point 3: Define every acronym in their first use in the manuscript
Response 2: For each acronym used for the first time in the text, such as "DFT+D" on page 2, "GGA+U" on page 3 has been defined in the corresponding first-time use position and is marked in red in the text. “h-BN” is already defined in the introduction section on page 1.
Point 4: The method is lacking. Please provide more step by step details on the theoretical methods applied and state clearly every assumption made.
Carefully read the Calculation methods, we present the relevant calculation details in the order of calculation and some explanations for the assumption, which are already marked in the text on page 2.
“The calculations implemented with CASTEP code based on density functional theory (DFT) [32-34] were performed to study structural, magnetic and electronic properties of 3d TM atoms (V, Cr, Mn, Fe, Co and Ni) adsorbed monolayer BC2N. The interelectron exchange correlation potential was used to describe the phase between electrons and ions by using Perdew –Wang 91 (PW91) [35] under the generalized gradient approximation (GGA). In order to avoid the interaction of adjacent TM atoms, we need to set the appropriate supercell size. The change of adsorption energy of 5×5 supercell and 4×4 supercell is less than 1%, it indicates that 4×4 supercell is large enough to avoid the interaction of adjacent TM atoms.
First, we completely relax the 4×4 BC2N supercell, the plane wave truncation energy is 500 eV. The Brillouin region integral of pure BC2N adopts the special k-point sampling method of 7 × 7 × 1 monkhost-pack to ensure the accuracy of the results, and the special k-point sampling of the monkhost-pack of the adsorption system is increased to 10×10×1 [37]. A 15Å vacuum space in the Z direction was applied to each side of the supercell to avoid the interactions between the individual layers. Grimme's semi-empirical dispersion correction (DFT+D) method is added to consider van der Waals interaction [36]. In all process of self-consistent calculation, the convergence criterion follows that the force on each atom is less than 0.05 eV/Å and the convergence threshold is 10-6 eV. The coulomb interaction (GGA+U) was introduced to describe the d electrons of TM atom [38].”
Reviewer 3 Report
The manuscript entitled “Structural, magnetic and electronic properties of 3d transition-metal atoms adsorbed monolayer BC2N: a first-principles study” reported the synthesis of BC2N and the adsorption of 3d transition metal atoms on the surface of the synthesized materials. The work has scientific and practical importance and the overall quality of the work presented in this manuscript meet the standard of the Journal. So, my recommendation is that with minor revision, the submitted article can be considered for publication in the journal of Materials. Just few comments and queries are below-
· Based on adsorption energy calculation, the authors concluded about the chemical or physical adsorption of the transition metals on the BC2N surface. Does the author do any other study to support the claim?
· Was there any study to see when the desorption of transition metal atoms takes place from the monolayer of BC2N surface which can give more insights about the material stability.
· Based on nanomagnetic properties, where the materials stand compared to other similar types of materials?
Author Response
Dear Reviewer:
Thank you for your comments concerning our manuscript entitled “Structural, magnetic and electronic properties of 3d transition-metal atoms adsorbed monolayer BC2N: a first-principles study” (ID: materials-501745). Those comments are all valuable and very helpful for revising and improving our paper, as well as the important guiding significance to our researches. We have studied comments carefully and have made correction which we hope meet with approval. Revised portion are marked in red in the paper.
Some corrections can be found in the manuscript.
Response to Reviewer 3 Comments
Point 1: Based on adsorption energy calculation, the authors concluded about the chemical or physical adsorption of the transition metals on the BC2N surface. Does the author do any other study to support the claim?
Response 1: Physical adsorption refers to adsorption between an adsorbent and an adsorbate by intermolecular attraction (ie, van der Waals force), and its adsorption energy is small (absolute value). Chemical adsorption refers to the chemical interaction between the adsorbent and the adsorbate, resulting in adsorption by chemical bonds, and the adsorption energy is large (absolute value).
By definition, we have determined the type of adsorption by the adsorption energy, whether the adsorbed atoms are bonded to BC2N can also determine the type of adsorption. By judging the adsorption energy, the adsorption energy of Cr and Mn atoms is weak (<0.5 ev) [42], which belongs to physical adsorption, and the adsorption energy is consistent with the corresponding references [43], [44].
From the perspective of whether the adsorbed atoms are bonded to BC2N, the charge density difference can be visually seen that the charge accumulation between Cr and Mn atoms and BC2N is the smallest. There is no clear channel between Mn (Cr) atoms and BC2N, it can be seen that it is difficult to form a bond between them, which belongs to physical adsorption. We did an analysis on page 6 about this part and marked it in red.
Theoretically, we know that the 3d orbital of free Cr, Mn atoms are half-filled. The half-filled 3d orbital weakens the interaction between TM atoms and monolayer BC2N, reducing the adsorption energies of these systems.
These studies can support our claim
Point 2: Was there any study to see when the desorption of transition metal atoms takes place from the monolayer of BC2N surface which can give more insights about the material stability.
Response 2:The stability of the material can be determined by the adsorption energy, binding strength. And the adsorption height between the adsorption atoms and BC2N can also reflect the stability of the adsorbent material. Some literature [1]-[3] also judges the stability of the adsorption system from these aspects. (1) The relative stability of this material is judged by comparing their adsorption energies on page 4. The small the adsorption energy, the more stable the adsorption. (2) The binding strength can further determine the structural stability and is analyzed on page 6. The stability increases with the binding strength. (3) The lower the adsorption height, the larger the absolute value of the adsorption energy from the side, and the more stable the adsorption, which can found from the table 1on page 5.
Our conclusions can be verified from these aspects.
[1] Li, L.; Zhang, H.; Cheng, X.; Miyamoto, Y. First-principles studies on 3d transition metal atom adsorbed twin graphene. Applied Surface Science. 2018, (https://doi.org/10.1016/j.apsusc.2018.02.075)
[2] Manadé, M.; Vi?Es, F.; Illas, F. Transition metal adatoms on graphene: a systematic density functional study. Carbon, 2015 95, S0008622315301895. (https://doi.org/10.1016/j.carbon.2015.08.072)
[3] Cao, C.; Wu, M.; Jiang, J.; Cheng, H.P. Transition metal adatom and dimer adsorbed on graphene: induced magnetization and electronic structures. Physical Review B Condensed Matter, 2010, 81(20), 2498-2502. (https://doi.org/10.1103/PhysRevB.81.205424)
Point 3: Based on nanomagnetic properties, where the materials stand compared to other similar types of materials?
The magnetic moments of transition metal atoms adsorbed on BC2N are larger than those adsorbed on InSe, so they are more sensitive to external magnetic fields and easier to use in spintronic applications and magnetic nanostructures. Data reference 41 citations.
41. Ju, W.; Li, T.; Zhou, Q.; Li, H.; Li, X.; Ma, D. Adsorption of 3d transition-metal atom on InSe monolayer: A first-principles study. Computational Materials Science, 2018, 150, 33-41. (https://doi.org/10.1016/j.commatsci.2018.03.067)